# Study on the Potential of Oil Spill Monitoring in a Port Environment Using Optical Reflectance

**Bikram Koirala** [1,*] , **Nicholus Mboga** [2,3] , **Robrecht Moelans** [4] , **Els Knaeps** [4] , **Seppe Sels** [2] , **Frederik Winters** [5] , **Svetlana Samsonova** [6] , **Steve Vanlanduit** [2] **and Paul Scheunders** [1]

1   Imec-Visionlab, University of Antwerp (CDE), Universiteitsplein 1,  2610 Antwerp, Belgium; paul.scheunders@uantwerpen.be
2   InViLab Research Group, University of Antwerp, Groenenborgerlaan 171, 2020 Antwerp, Belgium; nicholus.mboga@gim.be (N.M.); seppe.sels@uantwerpen.be (S.S.); steve.vanlanduit@uantwerpen.be (S.V.)
3   GIM NV Philipssite 5 Box 27, 3001 Leuven, Belgium
4   Vlaamse Instelling Voor Technologisch Onderzoek (VITO), Boeretang 200, 2400 Mol, Belgium; robrecht.moelans@vito.be (R.M.); els.knaeps@vito.be (E.K.)
5   RiskMatrix Group, Herkenrodesingel 4/1, 3500 Hasselt, Belgium; frederik.w@riskmatrix.be
6   Haven van Antwerpen-Brugge/Port of Antwerp-Bruges, Zaha Hadidplein 1, 2030 Antwerpen, Belgium; svetlana.samsonova@portofantwerpbruges.com
*   Correspondence: bikram.koirala@uantwerpen.be

**Abstract:** In this work, we studied the potential of the visible, near-infrared, and shortwave infrared wavelength regions for monitoring oil spill incidents using optical reflectance. First, a simple physical model was designed for accurate oil thickness and volume estimation using optical reflectance. The developed method was made invariant to changes in acquisition and illumination conditions. In the next step, an algorithm based on an artificial neural network was designed to detect spilled oil. The training samples that are required to optimize the parameters of the network were generated by utilizing the proposed physical model. To validate the method, experiments were conducted in laboratory and outdoor scenarios for detection and thickness/volume estimation on four different oil types. In particular, we developed hyperspectral datasets of oil samples with varying thickness between 500 μm and 5000 μm acquired using two different sensors, an Agrispec spectrometer and an Imec snapscan shortwave infrared hyperspectral camera, in strictly controlled experimental settings. To demonstrate the potential of the proposed method in outdoor environments using solely the visible wavelength region, we monitored the evolution of artificially spilled oil in an outdoor scene with an RGB camera mounted on a drone.

**Keywords:** hyperspectral; oil spill; multi-sensor dataset; RGB dataset

## 1. Introduction

Spilled oil has a significant impact on the environment, economy, and quality of life for inhabitants living near the spill location [1]. In marine environments, oil spills can occur due to ship discharge, leakage of oil pipelines, and unexpected disasters [2–4]. In order to respond to spilled oil properly, the oil has to be detected, and the thickness or volume of the oil spill has to be accurately estimated. A vast amount of literature exists on oil spill detection in oceans or coastal areas, based on remote sensing, usually with manned aircraft or satellites [5–8]. In [9], several remote sensing technologies and sensors currently being used to detect spilled oil [9] were reviewed. Passive microwave radiometers and laser fluorosensors were shown to be effective in detecting oil spills and estimating the thickness of spilled oil at sea [9].

The potential of optical remote sensing for oil spill monitoring has been studied. In [10], the spectral reflectance of five common oil types (crude oil, fuel oil, diesel oil, gasoline, and palm oil) was analyzed to test the potential of the optical remote sensing of

reflected sunlight in the visible, near-infrared, and shortwave infrared (VIS-NIR-SWIR) wavelength regions to detect different oil types involved in the spill. In [11], four supervised machine learning algorithms (random forest, support vector machine, deep neural network, and deep neural network with differential pooling) were utilized to identify different types of spilled oil. In [12], an algorithm was proposed to select an optimal three-band spectral index for classifying oil types. In [13], two different supervised machine learning algorithms (dense artificial neural networks and convolutional neural networks) were designed to predict the thickness of heavy fuel oil in hyperspectral images. In [14], the potential and limitations of optical remote sensing in detecting spilled oil were critically reviewed. Most of the common oil types have absorption features in the SWIR (1000 nm to 2500 nm) wavelength region around 1200 nm, 1400 nm, 1700 nm, and 2300 nm due to molecular combinations of C-H ([15]), $CH_2$, $CH_3$, or OH [16]. In the VIS (400 nm to 700 nm) and NIR (700 nm to 1000 nm) wavelength regions, most oil types do not have spectral signatures [9,14].

When oil is spilled, oil and water are left in a two-layer mixture or weathered to form emulsions. Oil spilled on the water surface initially spreads into a very thin layer [17]. Due to the high absorption power of water in the SWIR wavelength region, the spectral reflectance of the oil on top of the water is negligibly small. Unlike oil on top of water, emulsions reflect incident light diffusely. The reflected light contains features of CH [1,14], which are useful to quantitatively estimate the oil thickness/volume or concentration. In the remote sensing community, spectral reflectance of thick emulsions has been extensively studied in the SWIR wavelength region to generate maps of oil-to-water ratios [18,19], to estimate the concentration of oil [20], and to estimate the thickness of oil [19,21–23]. Because weathering and emulsification are dynamic processes [17], the major challenge is to track the changes in the physical and chemical properties of the oil during the estimation.

Ports are polluted by oil spills on a regular basis. For example, in 2019, 50 clean-up interventions were registered at Port of Antwerp-Bruges in Belgium. The average costs are between EUR 1 and 1.2 million yearly. Only about 40% of these costs are reimbursed by the offender. Solutions are required to accurately and quickly detect oil incidents in port areas. The literature on oil spill detection in a port environment is very scarce. Most available technologies are not directly applicable in a port environment. The specific case of the port environment adds complexity, which must be taken into account when selecting appropriate technology and algorithms:

- The port environment is much more cluttered. The presence of algae, sediments, debris, and infrastructure like vessels and docks leads to shadows, waves, etc. This complicates the interpretation of the images and can lead to false positives.
- The thickness and size of the oil spill are considerably smaller than in coastal or marine environments.
- The turbidity of water must be high enough to act as a diffusely reflecting surface. Unlike in the marine environment, turbidity is generally low in port environments.
- In port environments, oil spills are more likely to be shaped as oil on top of the water than as emulsions, since spills should be detected as fast as possible (within a few minutes) and the formation of emulsions takes time (from a few hours to a few days).

The current method of oil spill identification in ports is based on coincidence. When port authority officers visually notice an oil spill, a cleaning company is contacted. Early detection and determination of the oil incident (location, size, type of product, etc.) result in a much faster response to a calamity (hence a lower clean-up cost) and significantly improve intervention results. Remote sensing technology has not yet been applied in an operational setting in Belgium. The required technology, however, is there, in the form of compact hyperspectral cameras in the VIS-NIR (VNIR) and SWIR regions that can be mounted on drone platforms.

The contributions and novelties of this study are summarized as follows:

1. We propose a simple physical model to accurately estimate the thickness of oil samples. The estimated thickness can then further be used to estimate the total volume of an oil

spill together with the oil spill boundary estimates. The developed method is invariant to changes in sensor type and illumination conditions. Based on this physical model, we propose a method to detect spilled oil. For this, artificial spectra of oil samples with varying thickness are generated based on the model, and on these samples, a supervised machine learning algorithm is trained to detect spilled oil.

2.  To validate the proposed approach, we performed laboratory experiments on a comprehensive hyperspectral dataset of oil samples (diesel oil, lubrication oil, fuel oil, and hydraulic oil) with varying thickness between 500 μm and 5000 μm using a spectroradiometer. The same experiments were repeated (different samples) using a hyperspectral camera in the SWIR region.

3.  To simulate realistic oil spill situations in port environments, we performed an extensive experiment in outdoor settings.

The remainder of this article is organized as follows: In Section 2, the proposed physical model is elaborated. In Section 3, we describe the laboratory datasets and experimental results. In Section 4, we describe the dataset acquired in outdoor settings and the experimental results. Section 5 is devoted to a discussion, followed by a conclusion in Section 6.

## 2. Methodology

Transparent oil samples are non-diffusely reflecting materials and require a reflective background in order to measure their total reflectance. Let us denote the intensity of incident light on the surface of a sample by $I(\lambda)$ and the intensity of reflected light by $J(\lambda)$. The spectral reflectance is then given by $R(\lambda) = J/I$. We further denote the thickness of the oil by $X$. An infinitesimal layer $dx$ now absorbs an $adx$ portion of the light passing through it, with $a(\lambda)$ being the absorption coefficient of the oil under consideration at wavelength $\lambda$. When the incident light is reflected back after interacting with the background surface, the same layer further absorbs an $adx$ portion of the light. The following two differential equations describe these two phenomena:

$$di = a(\lambda)idx$$
$$dj = -a(\lambda)jdx \tag{1}$$

where $i$ and $j$ are the intensity values of incident light and reflected light, respectively, at an arbitrary depth $x$. One obtains

$$dj/j - di/i = dlogj - dlogi = dlog(j/i) = -2a(\lambda)dx \tag{2}$$

Integrating both sides of Equation (2) gives

$$\int_0^X dlog(j/i) = -2a(\lambda)\int_0^X dx$$
$$log(j/i) \mid_0^X = -2a(\lambda)X$$
$$log(J/I) - log(J'/I) = -2a(\lambda)X$$
$$log(R(\lambda)) - log(R'(\lambda)) = -2a(\lambda)X$$
$$R(\lambda) = R'(\lambda)exp(-2a(\lambda)X) \tag{3}$$

where $R'(\lambda)$ is the reflectance of the background surface. When assuming that the oil sample is non-scattering, the same equation can be derived from the Kubelka–Munk theory [24].

Estimating the thickness ($X$) of an oil sample from its spectral reflectance ($\mathbf{R} = [R(\lambda_1), \cdots, R(\lambda_d)]^T$) then boils down to inverting Equation (3), i.e., minimizing $\|\mathbf{R} - \mathbf{R}' \odot \exp(-2\mathbf{a}X)\|^2$, s.t. $X \geq 0$. Here, $d$ denotes the number of spectral bands, and $\odot$ is the elementwise multiplication of two vectors. For that, the spectral reflectance of the background surface ($\mathbf{R}'$) and the absorption coefficients ($\mathbf{a}$) of the oil sample are required. The above derivation explicitly assumes that both the incident angle and the reflection

angle are zero. When they deviate from zero, the following equation has to be minimized:
$\left\| \mathbf{R} - \mathbf{R}' \odot \exp(-(\sec(\theta_{inc}) + \sec(\theta_{ref}))\mathbf{a}X) \right\|^2$, s.t. $X \geq 0$, where $\theta_{inc}$ and $\theta_{inc}$ denote the
incident and reflection angles, respectively.

The absorption coefficients of an oil sample can be accurately measured with a spectrophotometer. Because each sensor (e.g., spectroradiometer or hyperspectral camera) has its own spectral response properties, the absorption coefficients measured by the spectrophotometer have to be corrected for sensor-specific effects by utilizing the spectral response function of the applied sensor.

In real-life scenarios, the measured spectral reflectance might be impaired by changes in acquisition conditions, such as illumination conditions, and distance and orientation with respect to the sensor. These effects mostly cause a random scaling effect in the measured reflectance spectra. In order to deal with this effect, each measured spectral reflectance value has to be projected onto the unit hypersphere (e.g., $\mathbf{R} = \mathbf{R}/\|\mathbf{R}\|$ and $\mathbf{R}' = \mathbf{R}'/\|\mathbf{R}'\|$).

Although the proposed method is designed to estimate the thickness of oil samples, it can also be utilized to detect spilled oil. For that, Equation (3) is used to simulate the reflectance spectra of oil samples with varying thickness. Training samples can then be generated through the augmentation of this simulated dataset with the spectral reflectance of the background water. A machine learning algorithm then learns a mapping between the input dataset and binary class labels (oil vs. no oil). In this work, we use a fully connected feed-forward neural network. This network has three layers: the input layer, a hidden layer with five nodes, and the output layer with two nodes. The hyperbolic tangent function $\left( \tanh(a) = \frac{\exp(a) - \exp(-a)}{\exp(a) + \exp(-a)} \right)$ is used as an activation function for the hidden layer, and the softmax activation function $\left( f(a_i) = \frac{\exp(a_i)}{\sum_{k=1}^{2} \exp(a_i)} \right)$, for the output layer. To train the network, we utilize the Levenberg–Marquardt backpropagation algorithm [25]. The training dataset is further split into a training subset and a validation subset. The cross-entropy loss is used to optimize the parameters of the network. The training subset is used to estimate the parameters of the network, while the validation subset is utilized to minimize the generalization error.

### 3. Laboratory Experiments

In this work, we investigated four different oil types (see Figure 1): diesel oil, lubrication oil (Smeerolie), fuel oil (Stookolie), and hydraulic oil. When an oil spill occurs in a port environment, typically, one of these four oil types is detected, unlike a marine environment where crude oil is typically spilled. Diesel oil is a mixture of saturated hydrocarbons and aromatic hydrocarbons. Lubrication oil is a mixture of petroleum hydrocarbon and lubricating additives. Fuel oil is a mixture of aliphatic and aromatic hydrocarbons, and hydraulic oil is a mixture of mineral oil and esters.

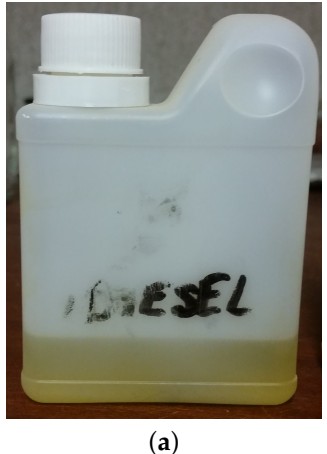
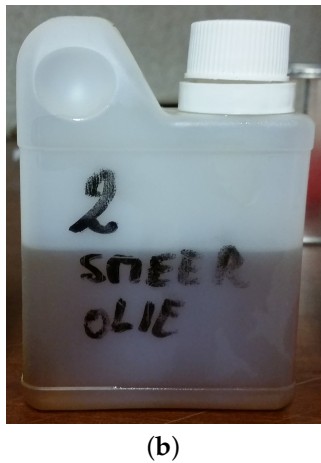

(**a**)   (**b**)

**Figure 1.** *Cont.*

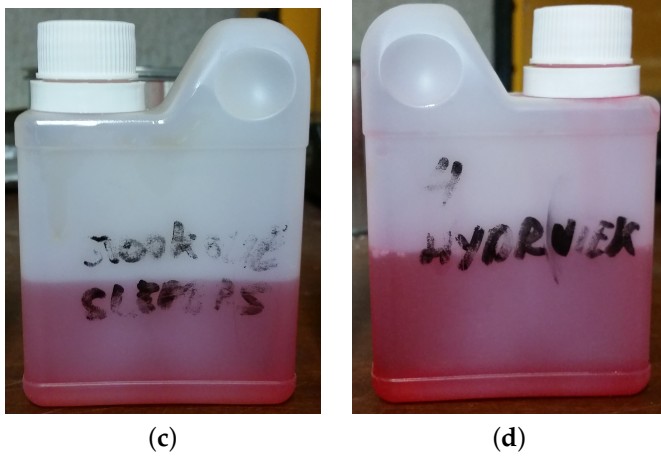

(**c**) 　　　　　　　　　　　　 (**d**)

**Figure 1.** The RGB images of four different oil types that were considered in this work. (**a**) Diesel oil;
(**b**) lubrication oil; (**c**) fuel oil; (**d**) hydraulic oil.

### 3.1. Measuring the Oil Absorption Spectra

In order to obtain the absorption spectra of all four oil types, the transmission spectra
of samples of approximately 2 mL were measured utilizing a spectrophotometer (200 nm to
2500 nm) (Cary 5000 UV-Vis-NIR Spectrophotometer, 5301 Stevens Creek Blvd, Santa Clara,
CA, USA. The absorption spectra of these oil types were obtained from the transmission
spectra (**T**) using the following formula: $\mathbf{a} = -\log(\mathbf{T})/\mathbf{X}$, where $X = 2$ mL. In Figure 2,
we show the absorption spectra (400 nm to 2200 nm) of the four oil types. These oil types
have almost the same absorption coefficient values in the SWIR wavelength region and
have spectral features around 1200 nm, 1400, and 1724 nm. These features indicate the
presence of carbon–hydrogen (1200 nm and 1700 nm) and oxygen–hydrogen (1400 nm)
bonds [1]. Since they visually appear differently colored, their absorption spectra differ
significantly in the visible wavelength region (400 nm to 700 nm). Given the fact that
there are three absorption peaks in the SWIR wavelength region, while there is only one
absorption peak in the visible wavelength region, the error in the estimated thickness of
the oil samples using the SWIR range is expected to be lower than that obtained using the
visible wavelength region.

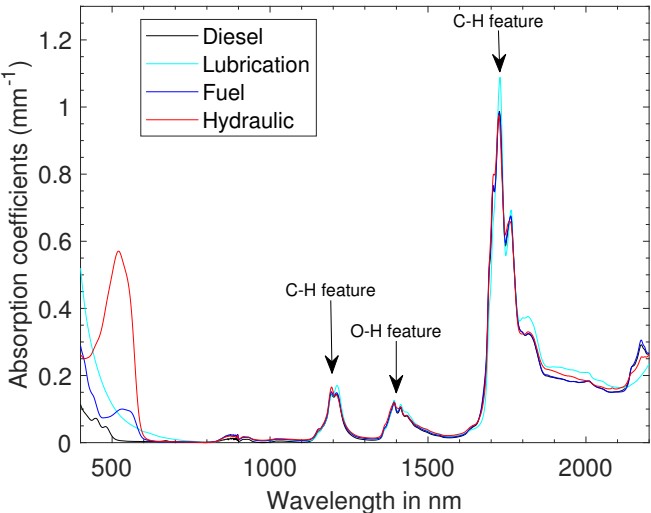

**Figure 2.** The absorption spectra of the four different oil types.

### 3.2. Oil Sample Preparation

For each oil type, 10 different samples were prepared with varying thickness between
500 μm and 5000 μm. These samples were placed inside white round sample holders with

an inner diameter of 20 mm, varying height between 0.5 and 5 mm, and edge thickness of approximately 3 mm. Although a dark sample holder as a background to spilled oil would be more suitable for simulating real-life scenarios, its high absorption capacity only allows thinner oil samples to be processed. To obtain reliable ground-truth thickness, the sample holders were completely filled. The sample mass was measured with a balance (Sartorius) with a resolution of 0.1 mg. To convert mass into volume, the density of each oil type was estimated by measuring the mass of a sample that occupied a volume of 100 mL. The thickness of the sample was then determined by dividing the volume of the sample by the area of the sample holder ($\pi \times$ inner diameter$^2$/4). The spectral reflectance of the samples was acquired using the ASD spectroradiometer, which has 2151 bands ranging from 350 nm to 2500 nm with a step size of 1 nm. We used an ASD Muglight as the illumination source in order to obtain spectra with a good signal-to-noise ratio. The illumination and acquisition angles were kept constant and were $35^0$ and $12^0$ for this sensor.

Since the completely filled sample holders experienced some loss during multiple transfers to and from the balance and sensor, sample preparation was repeated for measurement with a snapscan hyperspectral SWIR camera (manufactured by Imec, Leuven, Belgium). The dataset was acquired using the IMEC hyperspectral camera, which has 113 bands ranging from 1100 nm to 1670 nm. Four halogen lamps with diffusers were used for hemispherical–directional illumination to simulate uniform, real-world solar illumination. The acquisition angle was kept constant and was $0^0$ for this sensor. The distance between the sample and the camera was approximately 40 cm. Although the original frame size of the raw images was $100 \times 100$ pixels, we manually clipped $40 \times 40$ pixels from the center of the images to remove the edge of the sample holders. Since no spatial variation among the spectra was observed, the mean spectrum of the clipped image was considered for further analysis.

Figure 3 shows the spectra of the oil samples (approximately 5 mm thick) acquired with both the spectroradiometer and the hyperspectral camera. Although the absorption features of oil are visible in the spectra acquired with both sensors, the reflectance values are not exactly the same. This variability was caused by variations in illumination and acquisition angles and sensor differences.

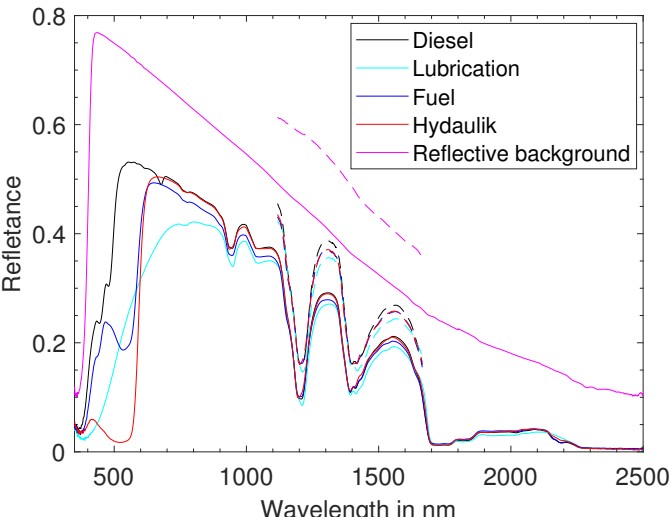

**Figure 3.** The spectral reflectance of 5 mm thick samples of four different oil types and a highly reflective sample holder acquired with the spectroradiometer (full line) and the hyperspectral camera (dashed line).

### 3.3. Experimental Results

The proposed thickness estimation method was validated on the datasets acquired with the spectroradiometer and the IMEC hyperspectral camera. A quantitative comparison

was conducted using the normalized absolute difference (NAD) between the estimated ($\hat{X}$) and the ground-truth thickness ($X$) values:

$$\text{NAD} = \frac{|X - \hat{X}|}{X} \times 100 \tag{4}$$

### 3.3.1. Estimating the Thickness of Oil with Hyperspectral Datasets

In the first experiment, the developed methodology was applied to the dataset acquired with the IMEC hyperspectral camera, i.e., the mean spectrum of the clipped image. For each oil type, the spectral response function of the hyperspectral camera, the spectral reflectance of the sample holder, and the absorption spectrum of the oil were utilized for inverting Equation (3). The results are shown in Figure 4.

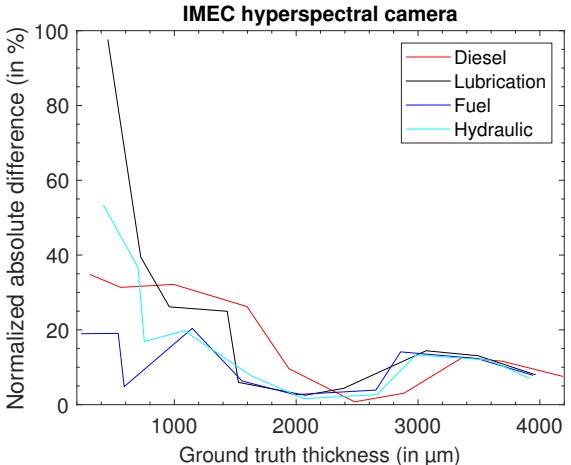

**Figure 4.** The NAD error as a function of thickness in the proposed method applied to the dataset acquired using the IMEC hyperspectral camera.

In the second experiment, the developed methodology was validated on the dataset acquired with the spectroradiometer. The method was applied separately for the VNIR (400–700 nm) and the SWIR (1118–1654 nm, to match the spectral range of the camera) regions. The results are shown in Figure 5.

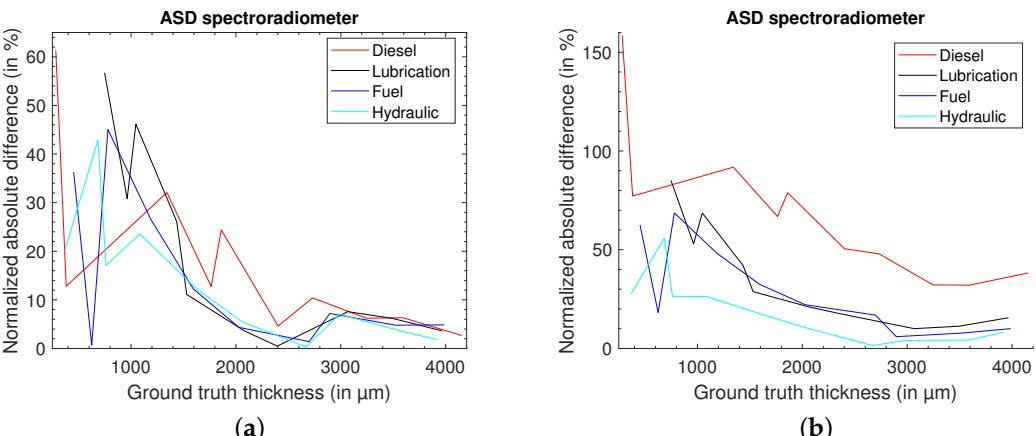

**Figure 5.** The NAD error as a function of the thickness in the proposed method applied to the dataset acquired using the spectroradiometer. (**a**) SWIR; (**b**) VNIR.

The outcomes of the experiments can be summarized as follows:

- The developed methodology was able to estimate the thickness of almost all oil samples from the datasets acquired in the SWIR wavelength region.

- For samples with thickness greater than 2000 µm, the error was 10% or less in the SWIR wavelength region. This demonstrates the potential of the proposed methodology.
- For samples with thickness lower than 2000 µm, the error was larger. This can be partially attributed to the uncertainty in the ground truth itself. While moving the samples during the measurement from the sensor location to the balance, small losses of oil regularly occurred. This introduced errors in the ground-truth thickness of the oil, especially for the thinner samples (thickness < 500 µm).
- As expected, the estimated thickness in the visible wavelength region was less accurate, especially for diesel oil. The best estimation in the visible wavelength region was that for hydraulic oil. This is due to the fact that it strongly absorbs incident light in the visible range (see Figure 2).

### 3.3.2. Simulating the Spectral Reflectance of the oil Samples in the Visible Wavelength Region

In the next experiment, we investigated the suitability of a multi-band camera in the visible range to estimate the thickness of oil samples. Because the oil samples that we investigated in this work do not have absorption features in the NIR wavelength region (see Figure 2), we only focused on the VIS wavelength region. For that, we chose two different multi-band cameras (MicaSense Dual Camera, 1300 N Northlake Way Suite 100, Seattle, WA, USA). Both cameras (Camera 1 and Camera 2) have three bands in the VIS wavelength region. We utilized the spectral response function of these cameras (see Figure 6) in order to convert the spectroradiometer dataset into three-band datasets.

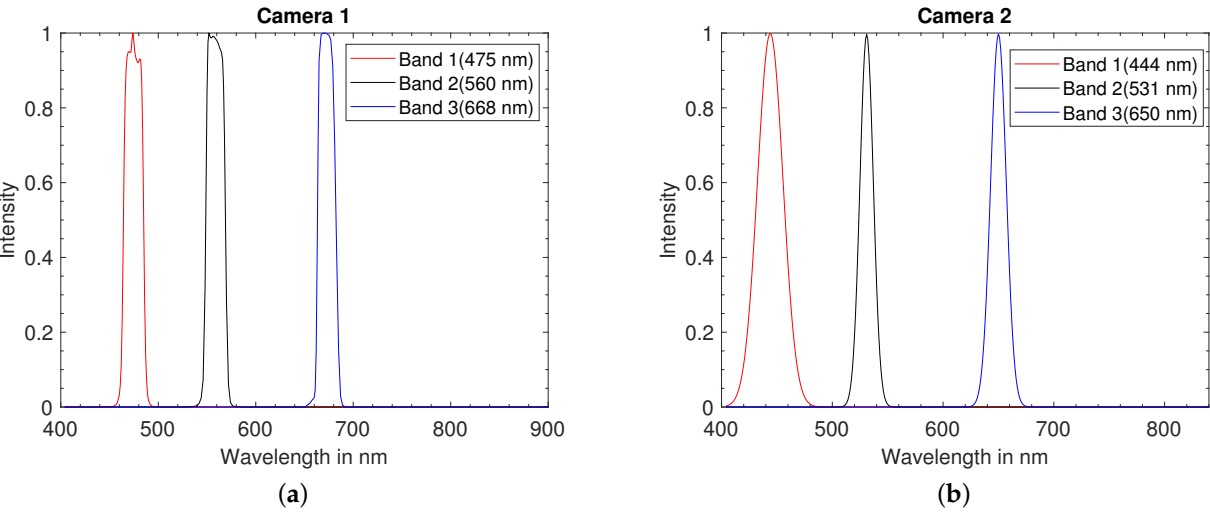

**Figure 6.** The spectral response of two multi-band cameras in the visible wavelength region. (**a**) Camera 1; (**b**) Camera 2.

In Figure 7, we show the estimated thickness of the oil samples from these three-band datasets. As expected, the results are comparable to the ones from the VNIR spectrophotometer dataset (see Figure 5). This can be explained by the fact that in the visible wavelength region, the absorption features of the oil types studied in this work are broad (see Figure 2). These results show the potential of using standard RGB cameras for oil spill detection and thickness measurements.

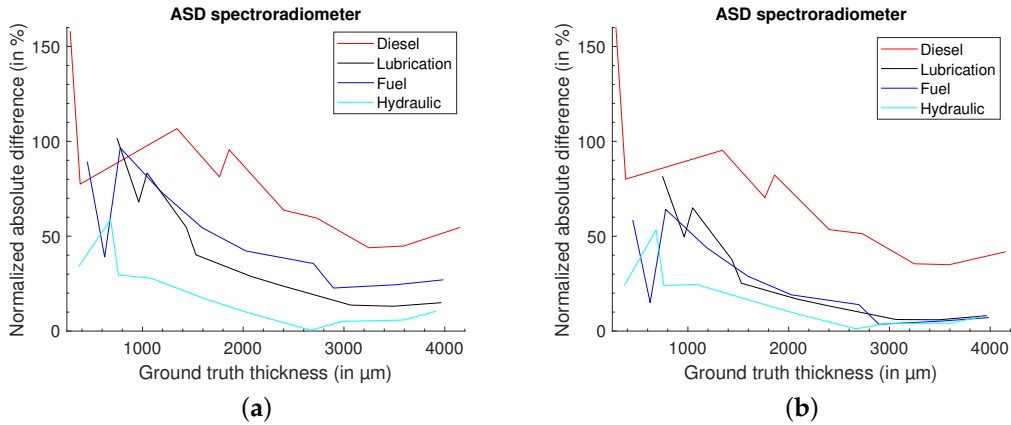

**Figure 7.** The NAD error as a function of the thickness in the proposed method applied to the three-band multi-spectral datasets. (**a**) Simulated Camera 1. (**b**) Simulated Camera 2.

### *3.4. Oil–Water Emulsions*

When spilled in the marine environment, oil forms emulsions due to turbulence from wind or human activities [1]. Emulsions are highly viscous and have distinct physical properties [1]. To analyze the spectral reflectance of oil–water emulsions, we produced emulsions by mixing oil with water. We fixed the ratio of oil to water to be 60:40. Each oil sample was placed in a transparent glass bottle, and a homogeneous emulsion mixture was produced by rotating the bottle for approximately 20 min. In Figure 8, we show RGB images of diesel oil and hydraulic oil, and their respective emulsions.

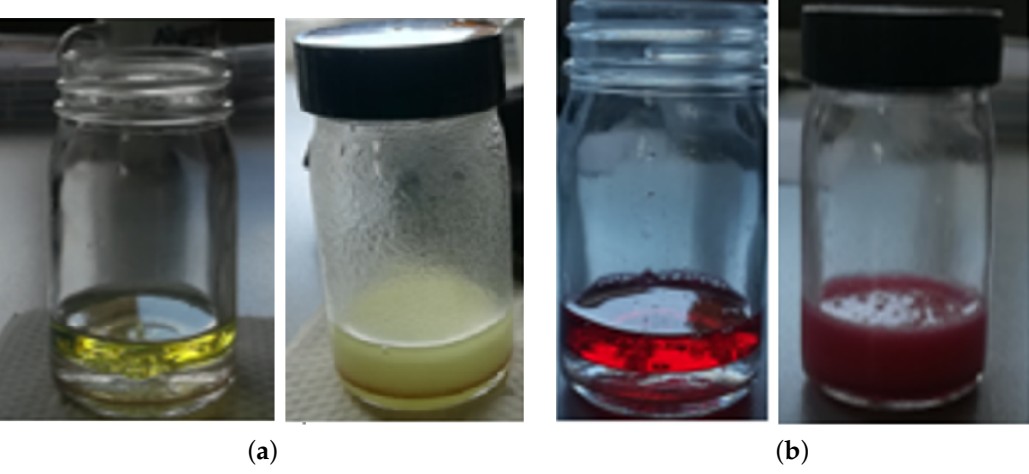

**Figure 8.** The RGB images of oil types and their respective emulsions. (**a**) Diesel oil; (**b**) hydraulic oil.

We placed these emulsions on top of tap water, and their spectral reflectance was acquired by utilizing a Specim FX17e hyperspectral camera, which has 224 bands ranging from 935 nm to 1702 nm. Although the original frame size of the raw images was 335 × 640 pixels, we manually clipped 40 × 40 pixels from the center of the emulsion. Since no spatial variation among the spectra was observed, the mean spectrum of the clipped image was considered for further analysis. In Figure 9, we show the spectral reflectance of the emulsions acquired with the Specim FX17e hyperspectral camera. As expected, the absorption features of the oil samples were visible in all emulsions. The reflectance of an emulsion is much higher than the reflectance of oil on top of water. In Section 4 (Figure 10), we study the spectral reflectance of oil on top of water under outdoor conditions. Since emulsions are unlikely to form in port environments, further investigation of emulsions is beyond the scope of this work.

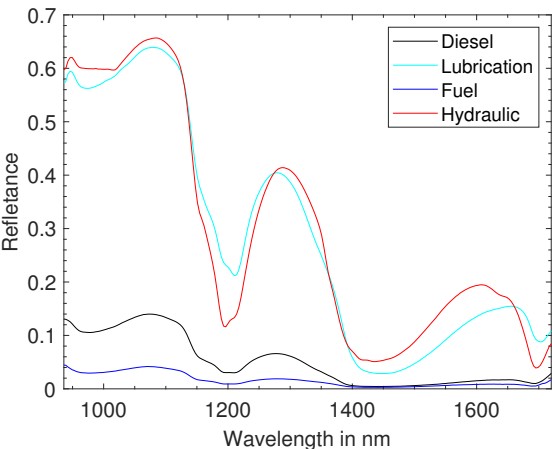

**Figure 9.** The spectral reflectance of four different emulsions on top of water acquired with a Specim FX17e hyperspectral camera.

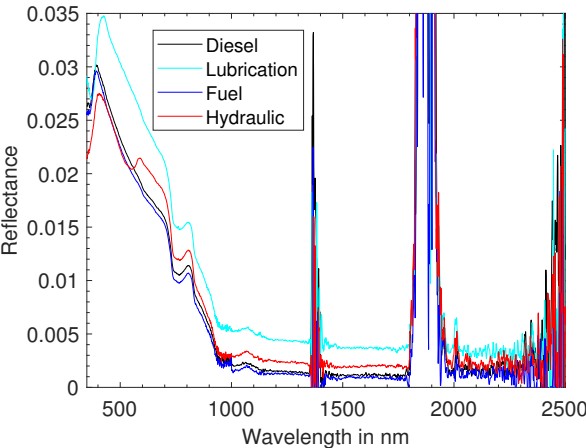

**Figure 10.** The spectral reflectance of four different oil types on top of the water was acquired with an ASD spectroradiometer.

## 4. Outdoor Experiment

To simulate realistic oil spill situations, we performed an experiment in outdoor settings on the 28 April 2022 around noon. The altitude of the sun was approximately $48^0$ from the horizon. At that time, the sky was open with some minor cirrus clouds at high altitudes. For this experiment, approximately 250 mL of oil from each oil type was spilled in a container filled with water. The size of the water container was 2 m $\times$ 2.7 m, and the depth of the water was 10 cm. The spilled oil mostly received direct sunlight. To minimize the light coming from the neighborhood building, the container was shielded with black plastic.

An ASD spectroradiometer was employed to acquire the spectral reflectance of the oil on top of the water. Figure 10 shows the reflectance of the four oil types. As expected, the spectral reflectance values were flat in the SWIR region due to the high absorption power of water (disregarding the atmospheric water absorption bands). However, in the visible range, the reflectance was approximately 3%. This allows for the use of RGB data for oil detection and volume estimation. Moreover, we have shown with the laboratory experiments in the previous section that it is possible to estimate thickness in the visible range (Figure 7).

We monitored the evolution of oil artificially spilled on water using an RGB camera mounted on a drone (see Figure 11). The altitude of the camera above the water container

was approximately 15 m. Hydraulic oil, lubrication oil, fuel oil, and diesel oil were spilled on the top left, top right, bottom left, and bottom right quadrants of the water container, respectively.

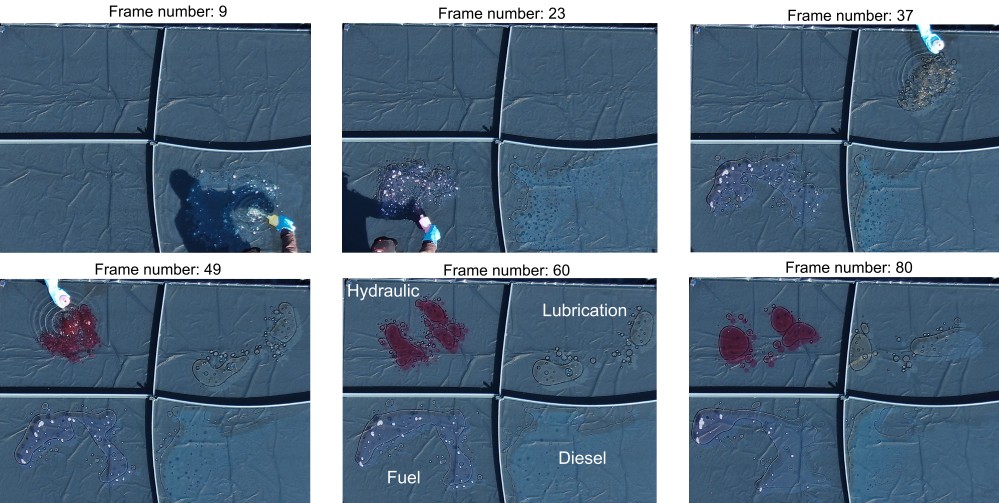

**Figure 11.** RGB images of oil spilled on top of 10 cm thick water. These images were acquired using an RGB camera (DJI) mounted on a drone.

*4.1. Detecting Oil in RGB Dataset*

In order to respond to spilled oil properly, the oil has to be detected. For this, we applied the proposed oil detection approach (see Section 2). We simulated 60,000 reflectance spectra of these four oil types in the VIS wavelength region (15,000 for each oil type). This simulated dataset was augmented with the background spectral reflectance (RGB image) of the water container (approximately 20,000 spectra). Then, we applied a fully connected feed-forward neural network to generate a binary classification map (i.e., oil vs. no oil). From the 80,000 spectra, approximately 56,000 spectra were used for training, while the remaining 16,000 spectra were used for validation. This learned model was further applied to the RGB dataset acquired using the DJI camera (see Figure 11). In Figure 12b, we show the oil spill detection map obtained for image (frame) number 60 (see Figure 12a). We also prepared a video of the oil spill detection map. This video is available at https: //github.com/VisionlabHyperspectral/Oil_spill/tree/main/Outdoor_experiment. As can be observed in Figure 12b, the proposed method accurately determined the spilled oil, and the boundary of the spilled oil is clearly visible in the oil spill detection map. Because the shiny (specular) backgrounds were not part of the training samples, the method classified them as spilled oil. These false positives can adversely affect the estimated volume of spilled oil, and an intelligent post-processing method is required to remove them. This is outside the scope of this work.

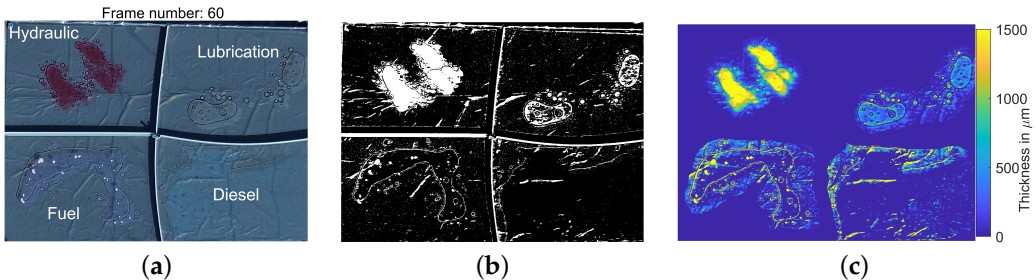

(**a**) (**b**) (**c**)

**Figure 12.** The obtained maps from image (frame) number 60. (**a**) RGB image; (**b**) oil spill detection map; (**c**) oil thickness map.

### 4.2. Estimating the Total Volume of Oil in RGB Dataset

To estimate the thickness of spilled oil, the proposed method requires prior information regarding the oil type and its absorption spectrum. The measured absorption spectra of these oil types in the laboratory settings were applied (see Figure 2). We resampled these spectra by utilizing the spectral response function of the applied sensor to match the number of bands and the spectral response of the RGB camera. In the next step, we manually cropped a region of interest for each oil type to accurately estimate its thickness. The mean spectrum of the water container was used as a background spectrum. Because the ground-truth thickness was not known, and only the total volume of the spill was known, we estimated the total volume of each oil type for a quantitative comparison. For that, the area covered by each pixel (approximately 2.2 mm × 2.2 mm) on the ground was multiplied by its estimated thickness. In Figure 13, we show the estimated volumes of the spilled oil as a function of the frame number. As can be observed, for all oil types, there was a positive bias in the estimated volume, i.e., the method initially estimated a positive oil volume when there was actually no oil yet in the scene (see the red line starting from image frame number 1). For diesel oil and fuel oil, the estimated volume was unreliable. This is due to the fact that these oil samples spread over a large area, resulting in an extremely thin layer of oil. Although the proposed method underestimated the volume of lubrication oil, the estimated volume was overall consistent in the image frames. This underestimation may result from thickness that is much lower than 2000 μm, the minimum required thickness for accurate estimation in the visible wavelength region. The volume of hydraulic oil was estimated reasonably well, demonstrating the potential of the proposed method to estimate oil thickness in outdoor scenarios.

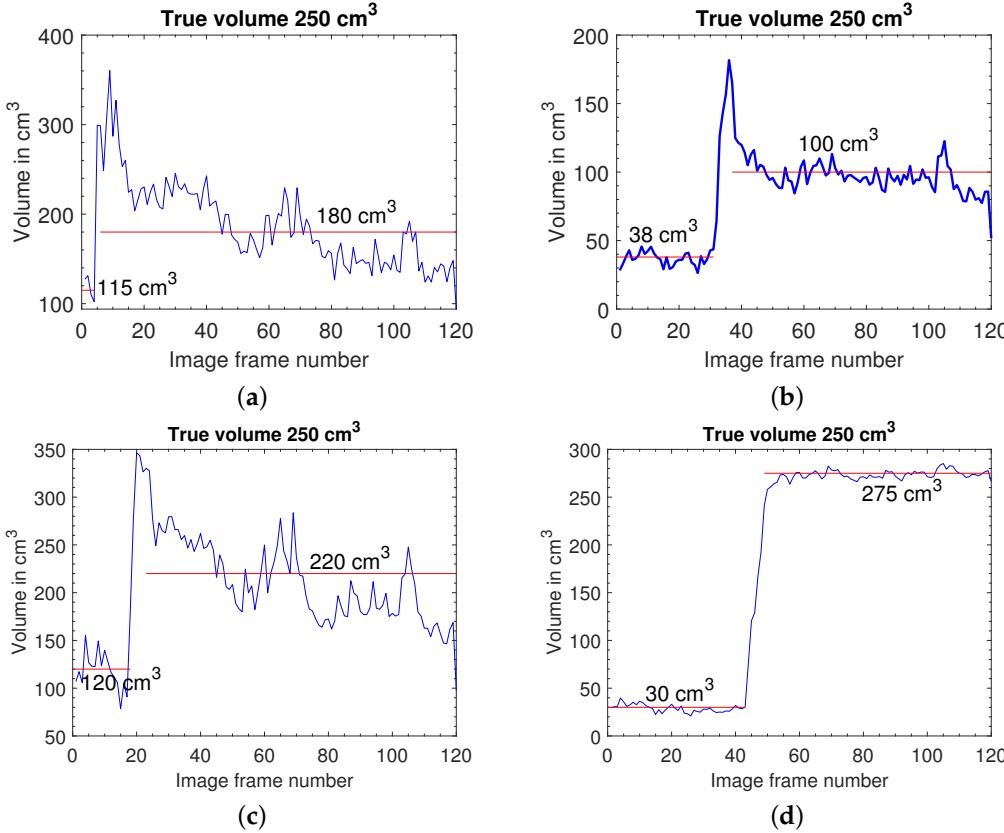

**Figure 13.** The estimated volumes of spilled oil. (**a**) Diesel oil; (**b**) lubrication oil; (**c**) fuel oil; (**d**) hydraulic oil. The positive bias in the estimated volume is shown by the red line (and its label) starting from frame number 1, while the average estimated volume is indicated by the second red line (and its label).

In Figure 12c, we show the estimated thickness maps obtained for image frame number 60 (Figure 12a), in which the spill was complete. Although the estimated thickness varied between 0 μm and 2500 μm, for better visualization, all pixels with thickness greater than 1500 μm are displayed as having thickness of 1500 μm. As can be observed, both hydraulic and lubrication oil spills were concentrated in a small region of the scene, resulting in better thickness estimation. The underestimation of the thickness of lubrication oil is due to the fact that its thickness is lower than 2000 μm, the minimum required thickness for accurate estimation (see Figure 7).

## 5. Discussion

Based on the literature and our own measurements and analyses, the following main conclusions are drawn:

- The thickness of oil samples can be accurately (NAD < 10%) estimated in the SWIR wavelength region only when the background surface is highly reflective. In real-life scenarios, oil is spilled on top of water. Because of the strong absorption power of water in the SWIR wavelength region, the intensity measured with a hyperspectral sensor is extremely low (see Figure 10) and does not allow the oil to be detected or the thickness of the oil to be estimated.

- The absorption features of oil can be observed in emulsions prepared by homogeneously mixing oil with water (see Figure 9). In port environments, the probability of emulsion formation due to the mixing of spilled oil and water has yet to be investigated. An interesting future research direction is the development of a methodology that can accurately estimate the thickness of oil from emulsions.

- The oil samples that were studied in this work have specific absorption features in the visible wavelength region. Although this is beneficial, due to the low absorption coefficient values of lubrication oil and fuel oil, we require at least 2000 μm thick oil samples to accurately (NAD < 20%) estimate their thickness. Since hydraulic oil has higher absorption coefficient values in the visible wavelength regions, thinner samples of this oil can be processed. As can be observed in Figures 5 and 7, the thickness of diesel oil cannot be accurately estimated in the visible wavelength region. Existing state-of-the-art methods utilize the specular properties of reflected light to infer the thickness of spilled oil. The major challenge for those methods is to differentiate between terrain spreading on water and oil. The proposed method, on the other hand, utilizes diffusely reflected light of the entire sample (oil + background) to estimate the thickness of spilled oil. The disadvantage of the proposed method is that it cannot accurately estimate the thickness of extremely thin oil samples.

- In outdoor scenarios, the acquired data may be impaired by inconsistent illumination conditions. Due to the normalization of the spectra and calibration with the spectral response functions of the sensors, the proposed methodology tackles this and can accurately estimate the volume of artificially spilled hydraulic oil (see Figure 13d) in outdoor scenarios. Although the estimated volume of lubrication oil was consistent for several RGB images (see Figure 13b), the developed methodology underestimated the total spilled volume. It seems that the thickness of this spilled oil type is lower than 2000 μm, the minimum required thickness to accurately estimate it.

- The proposed method requires the spectral reflectance of the background and information regarding the incident angle ($\theta_{inc}$) and the acquisition angle ($\theta_{ref}$). The spectral reflectance of the background can be generated by manually or automatically cropping a region in the scene where no oil spill occurs. On the other hand, $\theta_{inc}$ and $\theta_{ref}$ are obtained from the solar incident angle and the sensor's position and orientation.

- Recently, several algorithms have been developed for oil-type identification using the reflectance spectrum. In [12], an algorithm was proposed to optimize three-band spectral indices for differentiating oil types. Although these types of algorithms are suitable for diffusely reflecting samples, our oil samples are non-diffusely reflecting materials, and their absorption spectra and the spectral reflectance of the background

define the overall shape of the measured reflected light. From Figure 2, it is clear that all oil types have exactly the same absorption spectra in the shortwave infrared wavelength region, limiting the oil-species identification model to accurately differentiating them. However, in the visible wavelength region, the methodology proposed in [12] might be of help to differentiate different oil types according to their spectral reflectance. This will be the focus of our future work.

- Another interesting future research direction is to investigate the potential of the ultraviolet wavelength region to estimate the thickness of thinner oil samples.

The proposed method was developed in MATLAB and run on an Intel Core i9-12900KF CPU, 3.19 GHz machine with 16 cores. To estimate the thickness of spilled oil using the spectral reflectance, the proposed method required less than 40 milliseconds per spectrum. In order to train a fully connected feed-forward neural network, it took approximately 9 s, while the testing of the model was performed within 1 microsecond per spectrum.

## 6. Conclusions

In this paper, we propose a model to accurately simulate the spectral reflectance of oil lying on top of a reflective background and, based on this, to detect oil and estimate oil thickness in a port environment. Moreover, the proposed method is made independent of the changes in acquisition and illumination conditions. The proposed method was validated on datasets generated both in laboratory and outdoor settings and in cross-sensor situations. The proposed method not only accurately estimated the thickness of oil samples in laboratory settings but also accurately estimated the volume of thicker oil samples in outdoor scenarios from RGB images. The method was also found to be suitable to detect spilled oil. In future work, we will extend this method to accurately estimate the thickness of oil based on emulsions.

**Author Contributions:** Conceptualization, B.K., N.M., R.M., S.V. and P.S.; methodology, B.K. and P.S.; software: B.K.; validation, B.K. and N.M.; writing—original draft preparation, B.K.; writing—review and editing, N.M., R.M., E.K., S.S. (Seppe Sels), F.W., S.S. (Svetlana Samsonova), S.V. and P.S. All authors have read and agreed to the published version of the manuscript.

**Funding:** The research presented in this paper was funded by Research Foundation Flanders (project G031921N); Belgian Science Policy, Stereo IV program (project SR/00/400); and VLAIO, De Blauwe Cluster (project HBC.2021.0676).

**Data Availability Statement:** The datasets and the algorithm can be downloaded from the following link: https://github.com/VisionlabHyperspectral/Oil_spill.

**Conflicts of Interest:** The authors declare no conflict of interest.

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
