# Peer review of "Study on the Potential of Oil Spill Monitoring in a Port Environment Using Optical Reflectance"

_remotesensing, doi:10.3390/rs15204950_

Round 1
Reviewer 1 Report
Quantitative analysis of oil spills is a challenging issue. How to conduct rapid and real-time quantitative analysis is an unprecedented challenge. The authors attempt to conduct some works that may be useful in the field of oil spill remote sensing. However, there are some issues that need to be clarified in this paper. The detail comments are as follow:
1. The introduction part needs to be enriched with some recent studies on oil spill detection and identification using reflectance spectroscopy, which is include but not limited to the paper discussed in the following comments.
2. If there is only 1 subsection (1.1. Contributions and Novelties) in Section 1, there is no need to have a subsection title.
3. According the results of oil thickness inversion presented in Table 2 and Table 3, the error reduces as the oil thickness increases. This result is similar to a early study by Kieu and Law:
Hieu Trung Kieu & Adrian Wing-Keung Law (2022) Determination of surface film thickness of heavy fuel oil using hyperspectral imaging and deep neural networks, International Journal of Remote Sensing, 43:3, 997-1014, DOI: 10.1080/01431161.2022.2028200
However, the deep learning model proposed by Kieu and Low (2022) achieved good prediction for oil thicker than 50 um, but the model proposed by the authors still has considerable error at 600 um. The authors should consider comparing the proposed method with the deep learning model and illustrate the advantage of the proposed model.
4. What is the incident light condition and weather condition of the outdoor experiment? These factors should have significant influence on the measurement but is not explained in the paper.
5. The four types of oil used in the experiment are all refined light oil. Thus, the feasibility of the proposed model on heavy oil, such as crude oil and heavy diesel, which is commonly witnessed in oil spill event, is not tested.
6. Since the reflectance of oil spill is closely related to oil species, it is expect that the oil species need to be determined before calculating the oil thickness. There have been sufficient studies on this topic including:
M. Xie, S. Dong, T. Gou, Y. Li, B. Han, Evaluation and optimization of the three-band spectral indices for oil type identification using reflection spectrum, J. Quant. Spectrosc. Radiat. Trans. 304 (2023) 108609. https://doi.org/10.1016/j.jqsrt.2023.108609.
R.N. Clark, J.M. Curchin, T.M. Hoeffen, G.A. Swayze, Reflectance spectroscopy of organic compounds: 1. Alkanes, J. Geophys. Res. 114 (2009) E03001. https://doi.org/10.3390/s18010091.
J. Yang, J. Wan, Y. Ma, J. Zhang, and Y. Hu, “Characterization analysis and identification of common marine oil spill types using hyperspectral remote sensing,” Int. J. Remote Sens., vol. 41, no. 18, pp. 7163-7185, Jun. 2020.
The author should discuss how the proposed model could be integrated with an oil species identification model to form a more comprehensive oil spill monitoring method.
7. The reflectance in Fig. 3, 6, and 7 should have the unit of sr-1.
Reviewer 2 Report
Dear Authors,
Thank you for presenting the paper entitled "Study of the Potential of optical reflectance for oil spill monitoring in port environments".
You introduce excellently the problem you face, but the solution you propose is not clear.
You base your method on lightwave reflectance, and it is based on direct illumination. Despite the theoretical method description, standard RGB cameras can confuse terrain spread on the water with oil.
You should provide a reliable analysis to confirm you uniquely identify the oil.
At least, you should provide the computational cost of your method.
Best
Reviewer 3 Report
1 The method is not affected by the change of acquisition conditions and lighting conditions, and the experimental data in indoor and outdoor environments validate the method.
2 Consider whether it is also acceptable in dynamic water environments
3. Whether the method is affected by the distance, such as whether it is feasible to use it on the sea surface, and whether the effective monitoring range is considered in the use.
4. For emulsified oil sample experiment, can we consider a variety of different oil-water ratios to make the experimental data more abundant?
5. There are problems in the expression of some contents in the article.
6 The tabular data can be converted into a graph to show the difference
no
Round 2
Reviewer 1 Report
1. The title of the paper reads awkwardly. It is suggested to change it as "Study on the potential of oil spill monitoring in port environment using optical reflectance".
2. The last sentence in the abstract should be move to somewhere in Section 3 or 4, or separately as a data availability statement.
3. The sentence in L77 should be changed as "The contributions and novelties of this study are summarized as follow:" without bold fonts.
4. Once Comments 2 is addressed, the paragraph in L90 can be removed.
5. Given the fact that the proposed model focus on the refined light oil in port environment as the authors stated in the responses, the first sentence in the Conclusion part (L342) should be changed as "..., and from this to detect oil and to estimate oil thickness in port environment."
Reviewer 2 Report
Dear Authors,
Thank you for your work. You answered most questions. I suggest you put in evidence the method's limit using a clear sentence similar to the one you wrote in the question's reply.
We thank the reviewer for his/her accurate observation. If the estimation depends on the spilt oil's specular properties, RGB cameras could confuse terrain spread on the water with oil. Our method, however only utilizes diffusely reflected light of the entire sample (oil+background) to estimate the thickness
of spilled oil.
Best
